



# Unified observation co-existing volcanic sulphur dioxide and sulphate aerosols using ground-based Fourier transform infrared spectroscopy

Pasquale Sellitto[1], Henda Guermazi[1,2,3], Elisa Carboni[4], Richard Siddans[5], and Mike Burton[6]

[1]Laboratoire Inter-universitaire des Systèmes Atmosphériques, Université Paris-Est Créteil, France.
[2]Laboratoire de Météorologie Dynamique, École Normale Supérieure, Paris, France.
[3]National School of Engineers of Sfax, Water, Energy and Environment Laboratory L3E, University of Sfax, Tunisia.
[4]COMET, Atmospheric, Oceanic and Planetary Physics, University of Oxford, Clarendon Laboratory, Oxford, UK.
[5]UK Reseach and Innovation, Science and Technology Facilities Council, Rutherford Appleton Laboratory, Chilton, UK.
[6]School of Earth, Atmospheric and Environmental Sciences, University of Manchester, Manchester, UK.

**Correspondence:** Pasquale Sellitto (pasquale.sellitto@lisa.u-pec.fr)

**Abstract.** We developed an optimal-estimation algorithm to simultaneously retrieve, for the first time, co-emitted volcanic gaseous $SO_2$ and sulphate aerosols (SA) from ground-based FTIR observations. These effluents, both linked to magmatic/degassing and subsequent atmospheric evolution processes, have overlapping spectral signatures leading to mutual potential interferences when retrieving one species without considering the other. We show that significant overestimations may be introduced in $SO_2$

retrievals if the radiative impact of co-existent SA is not accounted for, which may have impacted existing SO2 long-term series, e.g. from satellite platform. The method was applied to proximal observations at Masaya volcano, where $SO_2$ and SA concentrations, and SA acidity were retrieved. A gas-to-particle sulphur partitioning of 400 and a strong SA acidity (sulphuric acid concentration: 65%) where found, consistently with past in-situ observations at this volcano. This method is easily exportable to other volcanoes, to monitor magma extraction processes and the atmospheric sulphur cycle.

## 1 Introduction

Volcanic gas and particulate emissions affect tropospheric and stratospheric compositions, air quality and the environment, the distribution and optical properties of low and high clouds, the Earth radiation budget from the regional to the global scale, and therefore climate (e.g., von Glasow et al., 2009; Robock, 2000). One of the most important environmental pollutants and the main source of radiative forcing from volcanoes are long-lived acidic and highly reflective sulfate aerosols (SA), directly

emitted (primary SA) or formed by gas-to-particle conversion of sulfur dioxide ($SO_2$) emissions (secondary SA).

Observing these volcanic emissions and their atmospheric processes and variability in space and time using ground-based and satellite remote sensing is a crucial step towards understanding and quantifying their environmental and climatic impacts. Proximal integrated observations of different and interacting gaseous and particulate volcanic effluents is also important to gain insights into magmatic degassing processes and eruption forecasting. Fourier Transform InfraRed (FTIR) spectroscopy

(e.g., Oppenheimer et al., 1998; Francis et al., 1998; Duffell et al., 2003; Burton et al., 2007) is a powerful tool in this context.



Ground-based FTIR is an ideal instrument for measuring magmatic degassing, as it allows remote quantification of the major magmatic gases, including $H_2O$, $CO_2$, $SO_2$, HCl, HF and also trace gases such as $SiF_4$, CO and OCS. Typically a radiation source is needed, and this may be an artificial infrared lamp or hot rock/lava (e.g., Allard et al., 2005). Correspondingly, open-path (OP) FTIR is based on an atmospheric path of a known distance between an artificial or natural source and an FTIR optic system. These spectrometers collect spectra in a large spectral range, which contains broad spectral features arising from volcanic aerosols as well as more finely structured molecular absorption signatures. However, until now, these broad-band aerosol features have not been utilised, as the research focus has been exclusively on magmatic gas emissions. This means a very rich resource of information on volcanic aerosols and their processes is potentially available in previously measured FTIR spectra from volcanoes world-wide.

The possibility of sporadic detection of SA, without specific physicochemical characterisation, from high-spectral-resolution infrared satellite instruments has been shown in the past, for relatively strong volcanic eruptions with stratospheric injection (Haywood et al., 2010; Karagulian et al., 2010). In this work, we demonstrate that systematic detection, quantification and chemical characterisation of volcanic sulphate aerosols using OP-FTIR spectrometry is feasible. As $SO_2$ and SA emission are inter-connected by magmatic/degassing processes, by the short-term/small-scale atmospheric processes (e.g., Sellitto et al., 2017a) and have absorption features in the same spectral range (e.g., Sellitto and Legras, 2016; Sellitto et al., 2017b), we explore the possibility of characterising simultaneously these two volcanic effluents. This co-retrieval is intended to limit mutual biases, which are expected if $SO_2$ and SA are retrieved individually (e.g., Guermazi et al., 2017). In addition, this co-retrieval provides, in principle, simultaneous information on two interacting species, contributing a constraint to the inner volcanic and atmospheric sulphur cycle. This new inversion method (described in Sect. 3) is then applied to OP-FTIR observations taken at Masaya volcano during a measurement campaign conducted in 1998 (described in Sect. 2). Results are given and discussed in Sect. 4 and conclusions are drawn in Sect. 5.

## 2 The case study: Masaya volcano and the observation geometry

Masaya volcano (11.98° N, 86.16° W) is located in the central American volcanic belt, which runs from Guatemala to Costa Rica, in the north-south direction. It is situated about 25 km south-east of Managua city, Nicaragua, with an elevation of about 600 m. It is structured as a basaltic-andesitic shield caldera. Masaya is one of the world's most persistent sources of passive magmatic degassing with a relatively stable degassing rate and episodic stronger events (Rymer et al., 1998). The active vent releases $SO_2$,whose fluxes range from about 500 to 2500 t/day (Duffell et al., 2003; Mather et al., 2006; de Moor et al., 2013; Carn et al., 2017). Masaya is, therefore, one of the largest contributors of volcanic gas emissions in the central American arc (de Moor et al., 2017). One of Masaya's most remarkable features is its accessibility; it is literally a drive in volcano, with a car park next to the main degassing crater. This means that it has been used as a natural laboratory to conduct investigations into magmatic degassing and aerosol evolution. Direct air masses sampling and sub-sequent laboratory analyses of Masaya's plume to investigate aerosol composition and burden was conducted during several campaigns (e.g., Allen et al., 2002; Martin et al., 2011).



The data used in this work were collected not with direct sampling but OP-FTIR remote sensing, during a bi-annual (1998-1999) measurement campaign. In particular, we focus on high quality spectra collected during a measurement session from about 16:40 to 17:10 UTC of 15/03/1998. During these measurements Masaya was not in an eruptive period and observations are therefore representative of the normal passive degassing activity. During the campaign, a MIDAC Corporation OP-FTIR

spectrometer was deployed on the top of Santiago crater. The observation geometry for this instrumental setup is depicted in Fig. 2 of (Horrocks et al., 1999): an infrared lamp and an FTIR spectrometer were placed on two sides of the active Masaya crater, so that the lamp's radiation is observed by the FTIR after it is transmitted through the plume, with a total distance of 518 m. More details on the FTIR system and its technical specifications, the campaign and its major results can be found in (Horrocks et al., 1999; Burton et al., 2000).

**3   Methodology**

With reference to Fig. 2 of Horrocks et al. (1999), the radiance spectrum measured by the spectrometer when the plume is in the line of sight is:

$$\mathrm{I}(\lambda) = \mathrm{I}_{\mathrm{lamp}}(\lambda)e^{-\tau_{\mathrm{tot}}(\lambda)} \qquad (1)$$

where $\mathrm{I}_{\mathrm{lamp}}(\lambda)$ is the radiance emitted by the lamp and the total optical depth $\tau_{tot}(\lambda) = \tau_{\mathrm{plume}}(\lambda) + \tau_{\mathrm{BG}}(\lambda)$ is the combination

of the total optical depth of the plume and of the background atmosphere. If an observation is taken when the plume is not in the line of sight of the spectrometer, a background reference spectrum is obtained:

$$\mathrm{I}_{\mathrm{BG}}(\lambda) = \mathrm{I}_{\mathrm{lamp}}(\lambda)e^{-\tau_{\mathrm{BG}}(\lambda)} \qquad (2)$$

By taking two observations with and without the plume in the line of sight, the plume optical depth can be isolated:

$$\frac{\mathrm{I}(\lambda)}{\mathrm{I}_{\mathrm{BG}(\lambda)}} = e^{-\tau_{\mathrm{plume}}(\lambda)} \qquad (3)$$

If the plume is considered as homogeneous, its optical depth can be expressed as follows:

$$\tau_{\mathrm{plume}}(\lambda) = \int\limits_{path} k_{\mathrm{plume}}^{\mathrm{ext}}(\lambda)dl = k_{\mathrm{plume}}^{\mathrm{ext}}(\lambda)L \qquad (4)$$

where $L$ is the path length into the plume and $k_{\mathrm{plume}}^{\mathrm{ext}}(\lambda)$ is the spectral extinction coefficient of the plume, which encompasses both absorption by volcanic gases and absorption and scattering by particles. To avoid radiative interferences with volcanic gaseous effluents other than $SO_2$, we: 1) restricted our analyses to the spectral range 800-1170 cm$^{-1}$, where only $SO_2$ (roto-

vibrational $\nu_1$ band (e.g., Carboni et al., 2012)) and water vapour have absorption bands, and 2) carefully selected spectral micro-windows within this interval to single out the spectral regions not affected by water vapour lines absorption. The water vapour continuum absorption, in this band, is very small and can be neglected (Shine et al., 2016). In the selected spectral micro-windows, the extinction coefficient of the plume can be expressed as follows:

$$k_{\mathrm{plume}}^{\mathrm{ext}}(\lambda) = k_{\mathrm{aer}}^{\mathrm{ext}}(\lambda) + k_{SO_2}^{\mathrm{abs}}(\lambda) \qquad (5)$$





In the previous equation, $k_{\mathrm{aer}}^{\mathrm{ext}}(\lambda)$ represents the extinction by particulate matter in the plume, i.e. sulphate aerosols, ash or condensed water. Ash emissions were not visually observed during the measurement session. In addition, the spectral signature of ash and condensed water, both very different and distinguishable from SA spectral signature, is not observed in our dataset, so we exclude the presence of both types of particles.

We first selected a background reference spectrum based on the least amount of hydrochloric acid (HCl) measured in the plume, with an independent method, as done by Horrocks et al. (1999). The HCl is abundant in the plume. Then, using Eq.s 3 and 4, we derived the measured $k_{\mathrm{plume}}^{\mathrm{ext}}(\lambda)$ as follows:

$$k_{\mathrm{plume,meas}}^{\mathrm{ext}}(\lambda) = \frac{1}{L} \ln \frac{I_{\mathrm{BG}}(\lambda)}{I(\lambda)} \tag{6}$$

Using an optimal estimation method based on the Levenberg-Marquardt minimisation algorithm (Rodgers, 2000, page 92-
93 and references therein) (more details on the set-up of this method are given in the Appendix), we fitted the measured $k_{\mathrm{plume,meas}}^{\mathrm{ext}}(\lambda)$ and a modeled $k_{\mathrm{plume,mod}}^{\mathrm{ext}}(\lambda)$, parameterised as follows:

$$k_{\mathrm{plume,mod}}^{\mathrm{ext}}(\lambda,[\mathrm{H_2SO_4}]) = \mathrm{M_{SA}^r}\, k_{\mathrm{SA}}^{\mathrm{ext}}(\lambda,[\mathrm{H_2SO_4}],\mathrm{M_{SA}}) + \rho_{\mathrm{SO_2}}\sigma_{\mathrm{SO_2}}{}^{\mathrm{abs}}(\lambda,T,p) \tag{7}$$

In Eq. 7, $k_{\mathrm{SA}}^{\mathrm{ext}}(\lambda,[\mathrm{H_2SO_4}],\mathrm{M_{SA}})$ is the extinction coefficient of a target sulphate aerosol (SA) layer, calculated using a Mie code (http://eodg.atm.ox.ac.uk/MIE/). We fixed the size distribution, as a mono-modal log-normal distribution with 0.2
$\mu$m mean radius, 30 particles/cm$^3$ number concentration and 1.86 $\mu$m width. The SA layers have been taken as a dispersion of spherical droplets of a binary systems solution, composed of water and sulphuric acid. The sulphuric acid mixing ratio- ([$\mathrm{H_2SO_4}$]) and temperature-dependent complex refractive indices of these droplets have been taken from Biermann et al. (Biermann et al., 2000), for the temperature $T$ of the plume, which was assumed to be atmospheric temperature. Each combination of the size distribution parameters and [$\mathrm{H_2SO_4}$] corresponds to a total sulphate aerosol mass concentration. In
the spectral region between about 800 and 1200 cm$^{-1}$, a clear spectral signature of SA has been found by Sellitto and Legras (2016). Sellitto and Legras have also shown that the mass concentration, through a shift of the total absorption signature, and [$\mathrm{H_2SO_4}$], through a modification of the shape of the spectral signature, mostly determine the extinction coefficient of sulphate aerosols layers, except for extreme values of the mean radius (e.g. mean radius bigger than 0.4-0.5 $\mu$m). In this latter case, the scattering component of the total extinction may become important, leading to a stronger dependence of $k_{\mathrm{plume,mod}}^{\mathrm{ext}}$ on the size
distribution. Extreme values of the mean radius are unlikely for these proximal observations. Based on these considerations, in the fitting procedure we adjust the total mass concentration of the sulphate aerosols in the plume, by adjusting a mass concentration ratio para meter $\mathrm{M_{SA}^r}$, defined as the ratio between the total mass concentration and the mass concentration associated to the Mie-calculated extinction coefficient $k_{\mathrm{SA}}^{\mathrm{ext}}(\lambda,[\mathrm{H_2SO_4}],\mathrm{M_{SA}})$. In addition, the optimal estimation is run multiple times, using each time a different value of [$\mathrm{H_2SO_4}$]. An optimal value of [$\mathrm{H_2SO_4}$] is selected basing on the deviation of the modeled
and measured $k_{\mathrm{plume}}^{\mathrm{ext}}$. As for the SO$_2$ component, $\sigma_{\mathrm{SO_2}}^{\mathrm{abs}}(\lambda,T,p)$, is the SO$_2$ absorption cross section calculated, at temperature $T$ and pressure $p$ of the plume, using HITRAN spectroscopic data (position, shape and intensity of the absorption lines in the selected spectral region). Then, the number concentration $\rho_{\mathrm{SO_2}}$ is adjusted during the spectral fitting.





From the SA mass concentration ratio $M^r_{SA}$ and the SO$_2$ density $\rho_{SO_2}$, the total mass concentrations of sulphate aerosols and sulphur dioxide, $M^T_{SA}$ and $M^T_{SO2}$ are derived. Finally, the vector of the estimated plume's parameters is tri-dimensional and is composed by the SO$_2$ and sulphate aerosol mass concentration $M^T_{SO2}$ and $M^T_{SA}$, and the sulphate aerosol acidity (in terms of their sulphuric acid mixing ratio [H$_2$SO$_4$]). The error content of the SO$_2$ and sulphate aerosol mass concentration can be derived using the Rodgers framework (Rodgers, 2000). These uncertainties are dominated by error components transferred from the spectral measurement radiometric noise and the information content correlations between these two retrieved parameters, and is typically smaller than 15% for both parameters.

## 4  Results and discussion

Figure 1a shows the measured $k^{\text{ext}}_{\text{plume,meas}}$ and modeled (after spectral fitting) extinction coefficient $k^{\text{ext}}_{\text{plume,mod}}$ for the Masaya volcano plume, as well as the spectral residuals, for one individual FTIR observation in our dataset. The raw FTIR observations for this spectrum are in the Supplementary Material. In Fig. 1b, the individual contributions of the SA layer and of the SO$_2$ to the fitted $k^{\text{ext}}_{\text{plume,mod}}$ are also shown. A clear spectral signature of SA is visible in $k^{\text{ext}}_{\text{plume,meas}}$, with the two peculiar absorption features (two combinations of bend/stretch vibrational modes) of the undissolved H$_2$SO$_4$ molecules et 870-920 cm$^{-1}$ and 1150-1200 cm$^{-1}$ and the symmetric stretch $\nu_1$ bisulphite ion vibrational mode around 1050 cm$^{-1}$ (Biermann et al., 2000; Sellitto and Legras, 2016). These components of $k^{\text{ext}}_{\text{plume,mod}}$ are isolated in the SA-only spectrum of Figure 1b. Comparing the two subfigures, it is also apparent how an SA-only plume cannot completely explain the spectral shape of the measured spectrum. Adding the SO$_2$ contribution, the spectrum is more satisfactorily fitted. The residuals are between near-0, for wavenumbers < 1080 cm$^{-1}$, and about $1.0 \cdot 10^{-6}$ cm$^{-1}$ (up to 20%), for longer wavenumbers. The residuals are consistently contained in the $\pm 1\sigma_{\text{residuals}}$ interval, except for wavelengths longer than 1100 cm$^{-1}$. The bigger residuals above 1100 cm$^{-1}$ can be due to a) the lack of fine scale structures of the Biermann et al. (2000) refractive indices for SA (which are provided at a coarse spectral resolution), b) uncertainties in the SO$_2$ spectral absorption coefficient transferred from uncertainties in temperature/pressure used in their calculation and/or c) spectral drifts in the FTIR observations.

During the measurement session mentioned in Sect. 2, 38 in-plume spectral measurements are taken. In most cases, 25 spectra (66% of the overall observations), co-existent SO$_2$ and SA are detected. For 5 spectra (13%) only SO$_2$ is detected. For the remaining 8 spectra (21%), no clear SA and SO$_2$ signals are detected. For these latter cases, the measured plume's extinction coefficient is very small and we suppose that the plume was not or only partially in the line-of-sight of the FTIR. The mean values of the total mass concentrations of SA and SO$_2$ and the mixing ratio of SA, calculated by averaging the mentioned 25 individual retrievals obtained during the measurement session, are summarised in Table 1. With an independent retrieval algorithm, using a different spectral range (between 2465 and 2550 cm$^{-1}$), Horrocks et al. (1999) have found a SO$_2$ concentration of 98.4 ± 20.0 mg/m$^3$, for the same observation set at Masaya. Our retrieved SO$_2$ average total mass concentration (153.7 ± 85.3 mg/m$^3$), even if slightly bigger, is consistent with the one obtained by Horrocks et al. (1999). The bigger value in our average estimation can be due to the presence of volcanic fluorine compounds that might be present in the plume but are neglected in the present study. The retrieved average sulphate aerosol mass concentration is 0.4 ± 0.2



**Figure 1.** (a) Measured (solid black line) and modeled/fitted (dotted grey line) extinction coefficient of the Masaya volcano plume for the observation of 15/03/1998 at 17:06 UTC. Residuals (modeled minus measured extinction coefficient) are also shown in red. Orange lines show the corresponding standard deviation $\pm 1\sigma_{\text{residuals}} = \sqrt{2}\sigma_{k^{\text{ext}}_{\text{plume,meas}}}$ interval (here, excluding specific systematic spectroscopic errors transmitted from the SA laboratory measurements, $\sigma_{k^{\text{ext}}_{\text{plume,mod}}}$ has been considered conservatively equal to $\sigma_{k^{\text{ext}}_{\text{plume,meas}}}$; details on the calculation of $\sigma_{k^{\text{ext}}_{\text{plume,meas}}}$ are given in the Appendix). (b) Individual components of the modeled extinction coefficient due to SA (dark blue line) and SO$_2$ (sky blue line).



**Table 1.** Mean SO$_2$ and SA mass concentrations, and mean sulphuric acid mixing ratio in SA, for our retrieval session at Masaya volcano, with associated standard deviations (due to their variability during the measurement session). These mean values are obtained by averaging 25 individual retrievals.

| $M^T_{SO_2}$ (mg/m$^3$) | $M^T_{SA}$ (mg/m$^3$) | [H$_2$SO$_4$] (%) |
| --- | --- | --- |
| $153.7 \pm 85.3$ | $0.4 \pm 0.2$ | $65 \pm 18$ |

mg/m$^3$. The SO$_2$/SA ratio is then about 400. Due to the proximal observations, these aerosols can be considered as primary emissions and are still not significantly processed by the interaction with the atmosphere. The average H$_2$SO$_4$ mixing ratio is $65 \pm 18\%$. This result reveals highly acidic sulphate aerosols as observed previously with direct sampling (Allen et al., 2002). These acidic aerosols have been shown to be formed by small particles ($< 2\mu$m), accounting up to 80% of the total mass of

aerosol emissions at this volcano. The acidity comes from the sulphuric acid and hydrofluoric acid rapid conversion to particles, initiated from the oxidation of SO$_2$ (Mather et al., 2004). Thus, one important source of systematic error in our method is the possible presence of hydrofluoric acid dissolved in the aerosols droplets, that has been neglected in our radiative calculations. By considering the average SO$_2$ and SA mass concentrations and the average mixing ratio, it results that the gas-to-particle partition of sulphur in the plume is about 400 which is very similar than what found, in 2002, at Masaya by Allen et al. (2002)

(about 450). In any case, the temporal evolution of the three parameters and the SO$_2$/SA ratio are very variable and different samples, taken at different times, can be significantly different. Masaya volcanic activity is very stable but the gas emission rates vary and, critically, the aerosol evolution is probably strongly dependent on highly variable atmospheric parameters such as relative humidity and temperature, as well as the plume age at the moment of detection which depends partly on wind velocity. Notwithstanding this variability, the average sulphur partitioning seem very stable at longer time scales.

We tested the possibility that SA extinction could interfere with SO2 retrievals by retrieving SO$_2$-only, with our optimal estimation method, in spectra where the co-existence of the two volcanic plume components is clear, like the one in Fig. 1. In this case, we obtain an overestimation of the SO$_2$ total mass concentration of nearly 50 %, with respect to the case of co-retrieval of SO2 and SA mass concentrations. Even if this is an extreme case and inversion methods can be developed to partially compensate broad-band biases, like the one introduced by aerosols in the sampled airmasses, this example show how the

radiative interferences between SO$_2$ and SA may introduce overestimations on these retrievals. This is particularly important for the observation of volcanic plumes, with a potentially substantial impact in the case of aged plumes. For stratospheric eruptions, rapid formation of secondary SA has been recently observed, e.g. for Kasatochi volcano (Alaska, USA, August 2008) (Krotkov et al., 2010) and Nabro volcano (Eritrea, June 2011) (Penning de Vries et al., 2014) eruptions, with SO$_2$ lifetimes as short as a few hours. For volcanic eruptions with injection at lower altitudes (in the troposphere) or in persistent passive degassing

regimes, the rapid gas and aqueous phase oxidation/nucleation of SO$_2$ can lead to radiatively active layers within a few hours after the initial SO$_2$ emission (Sellitto et al., 2016; Guermazi et al., 2017), with potentially systematic volcanic signature on the regional aerosol optical properties downwind (Sellitto et al., 2017c). In general, the lifetime of SO$_2$ and the inherent time-scales





of SA formation and evolution are complex and depend on various factors including the solar irradiation, humidity temperature, PH and the presence of oxidants (e.g., Eatough et al., 1997). Then, we recommend to either co-retrieve $SO_2$ and SA or to take explicitly into account these interferences when attempting to retrieve chemically/micro-physically/radiatively-interconnected pollutants. These considerations apply notably to satellite observations, where the spectroscopic inversion problem can be

even more severe than in the present ground-based OP-FTIR case, due to the strongly ill-posed problem linked to a longer atmospheric path and the inherent radiative transfer. We also highlight that to date the typical band used to quantify SO2 with ground-based OP-FTIR is that at 2500 cm$^{-1}$ which is less affected by the aerosol extinction (e.g, Horrocks et al., 1999).

## 5 Conclusions

We developed a retrieval algorithm to observe volcanic $SO_2$ and SA emissions using ground-based OP-FTIR spectrometry.

To the best of our knowledge, this is the first time that SO2 and SA are simultaneously characterised using this or other ground-based or satellite-based remote-sensing techniques. The retrieval is based on a non-linear least square fitting algorithm, minimizing the difference between modeled and measured volcanic plumes' spectral extinction. A spectral micro-windows selection was performed in order to avoid the interference with water vapor absorption and to optimise the spectral fitting. The absorption of $SO_2$ and the extinction of SA have been modeled by means of high-spectral-resolution T/p-dependent

spectroscopic data and a Mie code driven by state-of-the-art aerosol optical properties, respectively.

We applied this method to proximal ground-based FTIR observations at Masaya volcano. We retrieved $SO_2$ and SA total mass concentrations and SA mixing ratio (linked to particles acidity). Average concentrations of $SO_2$, $SO_2$/SA ratios and acidity of aerosols are consistent with previous observations at Masaya. A gas-to-particle partition of sulphur of about 400 is found and a strong acidity ($[H_2SO_4]$ of about 65%), which is very consistent with past observations at Masaya, pointing at

reasonable long-term stability of Masaya primary sulphate emissions. We also underline the importance of taking into account SA when attempting to retrieve volcanic $SO_2$ from ground-based or satellite remote-sensing instruments around the 1200 cm$^{-1}$ $SO_2$ band. Neglecting the rapid formation (or even primary emission) of SA may lead to significant overestimation of the $SO_2$ (in our case, up to about 50 %). This method can be fruitfully applied to the quite large library of previously collected FTIR spectra from volcanoes world-wide to constrain the magmatic/atmospheric processes that determine their sulphur emissions

and gas-to-particle partitioning.

*Data availability.* The FTIR spectra used to produce Fig. 1 are available as Supplementary Material. The whole dataset can be provided on demand (pasquale.sellitto@lisa.u-pec.fr)

## Appendix A: Optimal Estimation set-up

An ad-hoc optimal estimation retrieval method is used in this work to retrieve $M_{SA}^r$ and $\rho_{SO_2}$, that are used to obtain the final

tri-dimensional output vector composed by the $SO_2$ and sulphate aerosol mass concentration $M_{SO2}^T$ and $M_{SA}^T$, and the sulphate



aerosol acidity [$H_2SO_4$] (see Sect. 3). The method is based on the minimisation of the following cost function $J$:

$$J = (\mathbf{x} - \mathbf{x_a})^T \mathbf{S_a}^{-1}(\mathbf{x} - \mathbf{x_a}) + (\mathbf{k}^{\text{ext}}_{\text{plume,meas}} - \mathbf{k}^{\text{ext}}_{\text{plume,mod}}(\mathbf{x}))^T \mathbf{S_\epsilon}^{-1}(\mathbf{k}^{\text{ext}}_{\text{plume,meas}} - \mathbf{k}^{\text{ext}}_{\text{plume,mod}}(\mathbf{x})) \qquad (A1)$$

In the previous equation, $\mathbf{x} = [M^r_{SA}, \rho_{SO_2}]$ is the state vector to be retrieved, $\mathbf{x_a}$ is the a-priori vector for $\mathbf{x}$, with its associated covariance matrix $\mathbf{S_a}$ and $\mathbf{S_\epsilon}$ is the measurement error covariance matrix (referred to the measured extinction coefficient

$\mathbf{k}^{\text{ext}}_{\text{plume,meas}}$). The measured and modeled extinction coefficient $\mathbf{k}^{\text{ext}}_{\text{plume,meas}}$ and $\mathbf{k}^{\text{ext}}_{\text{plume,mod}}(\mathbf{x})$ are the same of Sect. 3, with their implicit wavelength dependence and represented as vector.

The a-priori values of $M^r_{SA}$ and $\rho_{SO_2}$, elements of $\mathbf{x_a}$, are both taken as zero. In the a-priori covariance matrix $\mathbf{S_a}$, the diagonal elements describe the expected variability of the retrieved parameters, while off-diagonal elements express the possible co-variances: in this work only diagonal element are assigned non-zero values. The measurement noise covariance matrix

$\mathbf{S_\epsilon}$ has been constructed to represent the error measurements on the measured extinction coefficient. Thus, for each diagonal element (measurement noise transferred to the extinction coefficient at a given wavelength), we have considered the following expression, with reference to Eq. 6:

$$\sigma^2_{\text{k}_{\text{meas}}} = \frac{1}{L^2} \left( \frac{\sigma^2_{\text{I}_{\text{BG}}}}{\text{I}^2_{\text{BG}}} + \frac{\sigma^2_{\text{I}}}{\text{I}^2} \right) \qquad (A2)$$

Based on a conservative choice of a relative error of 1% for both background and in-plume observations, we obtain a $\sigma^2_{\text{k}_{\text{meas}}}$

variance of about $8.0 \cdot 10^{-14}$ cm$^{-2}$. Off-diagonal values of $\mathbf{S_\epsilon}$ are taken as zero.

*Author contributions.* P.S. and E.C conceived the method. P.S. and H.G. implemented the method and realised the inversions. R.S. developed the optimal-estimation software and E.C, H.G and P.S. contributed and adapted the software to the case-study. M.B. collected and pre-processed the FTIR measurements at Masaya volcano. All authors discussed the results and contributed to the final manuscript.

*Competing interests.* The authors declare that they have no conflict of interest.

*Acknowledgements.* This work has been supported by CNES under the project TOSCA/IASI, by the EC 7th Framework Program under grant 603557 (StratoClim) and by the ANR under the grant TTL-Xing. Part of the work has been performed in the context of the INGV SMED project. Giuseppe Salerno and Alessandro La Spina are gratefully acknowledged for the discussions on sulphate aerosols inversion from FTIR measurements.



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
