# Peer review of "Unified quantitative observation of co-existing volcanic sulphur dioxide and sulphate aerosols using ground-based Fourier transform infrared spectroscopy"

_Atmospheric Measurement Techniques, 2019_

## Referee Comment (RC1) · Michael Fromm (Referee) · 24 Jun 2019

Review of Sellitto et al., Unified observation co-existing volcanic sulphur dioxide and sulphate aerosols using ground-based Fourier transform infrared spectroscopy. Hereafter Sellitto at al. are referred to as "auth."

Reviewer: Mike Fromm

Auth have presented an analysis of high spectral-resolution IR data from a ground-based setup in close proximity to venting emissions from Masaya volcano. Their aim is to quantify abundances and composition of a volcanic plume that exhibits combi-

nations of SO2 gas and sulphate particles. In doing so they find that the co-retrieval reveals the potential for substantial systematic errors in stand-alone retrieval of SO2 when sulphates are mingled in the plume. The implications for and applications to other similar data sets (both ground-based and satellite-based) and volcanic events are made evident by auth. Their findings are a first and merit consideration in AMT.

The manuscript is very well written. It is a model for high-quality science reporting. It is concise. Its organization is logical, argumentation is clear and meaningful, and the analysis is robust. Auth fairly describe the parameters and the uncertainties involved. And they make the important point that many volcanic plumes need to be assessed by considering the co-presence of sulphate aerosol and SO2.

I have only a few minor and technical concerns, elaborated on below. It seems to me that the title doesn't adequately capture the over-arching thrust of the research. The title only refers to "observations" when actually the value of this work lies in the quantification of properties sensed by the observations of co-present sulphate particles and SO2 gas. If I am on the right track, I would encourage auth to revise title accordingly.

In case auth reject this suggestion, it is still necessary to amend the title by inserting "of" between "observation" and "co-existing".

Would auth be able to provide or cite a photograph of the plume that was analyzed? If there are SA particles in the plume, a photo cementing the idea of a sulfate feature would be strategic.

P2, L31. "Direct air masses sampling" is awkward to my eye. "Direct" seems to suggest that there may be a form of sampling that is "indirect," which is hard to understand. "In situ" may be clearer if that is what auth have in mind. Suggestion for rewording: "In situ air-mass sampling and subsequent laboratory analysis..."

Technical Issues:

Figure 1. . Label each panel with "a)" and "b)" correspondingly with the caption.

Figure 1b. The most visible difference between the lines for "Measured" versus "Modeled- SSA-Only" is solid versus dashed. The legend guides the reader to look for black versus dark blue; that color difference is difficult to see. Please consider changing dashed line to a gray shade as in 1a.

Figure 1b. In the legend, change "SSA" to "SA".

[Figure]

---

## Referee Comment (RC2) · Anonymous Referee #2 · 1 Jul 2019

The manuscript (MS) introduces a new method to simultaneously retrieve the co-existing volcanogenic SO2 and sulfate aerosols (SA) from ground-based FTIR observations. Data are collected from Masaya volcano during a bi-annual (1998- 1999) measurement campaign. Based on Mie calculations and using the non-linear least square fitting algorithm, the total mass concentration of the SA and SO2 (and their ratios) are derived that are consistent with previous observations at Masaya. The results show that ignoring co-existent SA can lead to substantial errors in SO2 estimations. This has very important implications for remote sensing of volcanic plumes.

The MS is very well structured and written. Methods and assumptions are clearly

explained. From my perspective, it is a significant contribution to volcanic plume observation and thus, should be published at AMT after addressing the following points.

Comments:

1- The term "co-emitted" is misleading. How can you make sure that H2SO4 is directly emitted during degassing? There are several studies that show high-temperature oxidation of H2S and SO2 affect the sulphur speciation in the plume (e.g. Martin et al 2006; Hoshyaripour et al 2012). Moreover, in ash-free plumes in the troposphere OH could oxidize the SO2 relatively quickly. Please replace this term with "co-existing".

2- P4.L2-4: The authors state "the spectral signature of ash and condensed water, both very different and distinguishable from SA spectral signature, is not observed in our dataset, so we exclude the presence of both types of particles". This makes me wonder:

2-1- What is the source of pre-existing particles on which H2SO4 condenses? If there is no water, why there is a 65% sulfate solution? This even contradicts the assumptions made later in the Mie calculations (binary solution).

2-2- What if there is a lot of ash particles and/or water droplets in the plume? Does this method work with ash- and droplet-free plumes (I seriously doubt if the second one exists in nature) only? If so, this statement should be revised: "This method is easily exportable to other volcanoes, to monitor magma extraction processes and the atmospheric sulphur cycle"

3- The authors claim that the method is "easily exportable to other volcanoes" but have analyzed only 30 minutes of data from 2-years measurements conducted 2 decades ago. What is specific about the data that makes it "high quality spectra" and how likely is it to get such data elsewhere? This will elaborate on the requirements and limitations of the method. It would be great to see a second example that shows the applicability of the method to other eruptions/volcanoes.

4- This MS is submitted to the special issue "StratoClim stratospheric and upper tropospheric processes for better climate predictions". Please explain how the results obtained from passively degassing volcano like Masaya (with tropospheric plumes) can be generalized and used for UTLS studies. Is it directly exportable to UTLS plumes? If not, what are the key aspects to consider?

References:

Hoshyaripour, G., M. Hort, and B. Langmann (2012), How does the hot core of a volcanic plume control the sulfur speciation in volcanic emission?, Geochem. Geophys. Geosyst., 13, Q07004, doi:10.1029/2011GC004020.

Martin, R. S., T. A. Mather, and D. M. Pyle (2006), High-temperature mixtures of magmatic and atmospheric gases, Geochem. Geophys. Geosyst., 7, Q04006, doi:10.1029/2005GC001186

---

## Author Comment (AC1) · 6 Sep 2019

Dear Referee,

please find our point-to-point reply to your comments in the Supplement.

Thank you for your constructive comments and suggestions, that helped improving our manuscript.

Pasquale Sellitto on behalf of all co-authors.

Please also note the supplement to this comment:

[Figure]

https://www.atmos-meas-tech-discuss.net/amt-2019-186/amt-2019-186-AC1-supplement.pdf

---

## Author Comment (AC2) · 6 Sep 2019

**Reply to Reviewers for the manuscript: "Unified observation of co-existing volcanic sulphur dioxide and sulphate aerosols using ground-based Fourier transform infrared spectroscopy", P. Sellitto et al.**

*Dear Editor, dear Reviewers,*

*Please find our reply to the comments of the two Reviewers (Mike Fromm and an anonymous Reviewer) for our manuscript ''Unified observation of co-existing volcanic sulphur dioxide and sulphate aerosols using ground-based Fourier transform infrared spectroscopy'', submitted for evaluation to AMT. All comments have been tackled and please find a point-by-point reply in the following.*

*Sincerely,*
*Pasquale Sellitto on behalf of all co-authors*

**Reviewer #1 (Mike Fromm)**

**Summary:**

Auth have presented an analysis of high spectral-resolution IR data from a ground-based setup in close proximity to venting emissions from Masaya volcano. Their aim is to quantify abundances and composition of a volcanic plume that exhibits combinations of SO2 gas and sulphate particles. In doing so they find that the co-retrieval reveals the potential for substantial systematic errors in stand-alone retrieval of SO2 when sulphates are mingled in the plume. The implications for and applications to other similar data sets (both ground-based and satellite-based) and volcanic events are made evident by auth. Their findings are a first and merit consideration in AMT. The manuscript is very well written. It is a model for high-quality science reporting. It is concise. Its organization is logical, argumentation is clear and meaningful, and the analysis is robust. Auth fairly describe the parameters and the uncertainties involved. And they make the important point that many volcanic plumes need to be assessed by considering the co-presence of sulphate aerosol and SO2.

*We would like to thank Mike Fromm for the kind words about our work and our effort of synthesis and clarity.*

I have only a few minor and technical concerns, elaborated on below.

**Minor Comments:**

MC1) It seems to me that the title doesn't adequately capture the over-arching thrust of the research. The title only refers to "observations" when actually the value of this work lies in the quantification of properties sensed by the observations of co-present sulphate particles and SO2 gas. If I am on the right track, I would encourage auth to revise title accordingly. In case auth reject this suggestion, it is still necessary to amend the title by inserting ''of'' between "observation" and "co-existing".

*We understand the concern of Mike Fromm and we propose the following new title: ''Unified quantitative observation of co-existing volcanic sulphur dioxide and sulphate aerosols using ground-based Fourier transform infrared spectroscopy''.*

MC2) Would auth be able to provide or cite a photograph of the plume that was analyzed? If there are SA particles in the plume, a photo cementing the idea of a sulfate feature would be strategic.

*A photo taken during the measurement campaign at Masaya, the case study presented in this*

*manuscript, has been already provided by Allen et al. (2002) (referenced in our manuscript). Please find the photo, with its original caption, in the picture on the right. Please note that the Open Discussion of an AMT paper is an integral part of such publication: thus, we prefer to add this picture here, within our comment, than directly in the manuscript. Please also note that Allen et al. (2002) report, based on*

[Figure]

Figure 1. (a) Aerial photograph of Masaya showing Santiago crater (in center) and sampling site (marked with a star). Image from the Instituto Nicaragüense de Estudios Territoriales (INETER). (b) Photograph of Santiago crater seen from above the east rim (with the sampling site indicated as in (a) and plume rising passively in center). (c) View into Santiago crater from sampling site showing the two degassing vents. Vent 2 formed in a phreatic explosion on 23 April 2001. (d) Close-up view of the 23 April 2001 vent, estimated to have a diameter of approximately 10 m.

*in-situ observation, the presence of highly acidic SA particles in the same plume sampled by FTIR and analysed in our manuscript: "The aerosols were highly acidic, with estimated pH of <1.0 in the fine aerosols. Sulfate was present mainly in smaller particles, with the fine fraction accounting for 80% of the mass."*

MC3) P2, L31. "Direct air masses sampling" is awkward to my eye. "Direct" seems to suggest that there may be a form of sampling that is "indirect," which is hard to understand. "In-situ" may be clearer if that is what auth have in mind. Suggestion for rewording: "In-situ air-mass sampling and subsequent laboratory analysis…"

*We accept the suggestion and have modified the text accordingly.*

**Technical Comments**

TC1) Figure 1. Label each panel with "a)" and "b)" correspondingly with the caption.

*Done.*

TC2) Figure 1b. The most visible difference between the lines for "Measured" versus "Modeled- SSA-Only" is solid versus dashed. The legend guides the reader to look for black versus dark blue; that color difference is difficult to see. Please consider changing dashed line to a gray shade as in 1a.

*We rather changed the dark blue dotted line to a lighter shade of blue and a dashed line. This should improve readability yet conserving the "blue" colour-code for individual signature components.*

TC3) Figure 1b. In the legend, change "SSA" to "SA".

*Done.*

**Reviewer #2 (Anonymous)**

**Summary:**

The manuscript (MS) introduces a new method to simultaneously retrieve the coexisting volcanogenic SO2 and sulfate aerosols (SA) from ground-based FTIR observations. Data are collected from Masaya volcano during a bi-annual (1998- 1999) measurement campaign. Based on Mie calculations and using the non-linear least square fitting algorithm, the total mass concentration of the SA and SO2 (and their ratios) are derived that are consistent with previous observations at Masaya. The results show that ignoring co-existent SA can lead to substantial errors in SO2 estimations. This has very important implications for remote sensing of volcanic plumes. The MS is very well structured and written. Methods and assumptions are clearly explained. From my perspective, it is a significant contribution to volcanic plume observation and thus, should be published at AMT after addressing the following points.

***Thank you very much for the kind words about our work.***

**Minor Comments:**

MC1) The term "co-emitted" is misleading. How can you make sure that H2SO4 is directly emitted during degassing? There are several studies that show high-temperature oxidation of H2S and SO2 affect the sulphur speciation in the plume (e.g. Martin et al 2006; Hoshyaripour et al 2012). Moreover, in ash-free plumes in the troposphere OH could oxidize the SO2 relatively quickly. Please replace this term with "co-existing".

***We fully agree and changed the text accordingly.***

MC2) P4.L2-4: The authors state "the spectral signature of ash and condensed water, both very different and distinguishable from SA spectral signature, is not observed in our dataset, so we exclude the presence of both types of particles". This makes me wonder: What is the source of pre-existing particles on which H2SO4 condenses? If there is no water, why there is a 65% sulfate solution? This even contradicts the assumptions made later in the Mie calculations (binary solution).

***The Reviewer #2 is right, the formulation of this sentence is misleading and we slightly reworded it: "The spectral signature of ash and pure condensed water, both very different and distinguishable from SA spectral signature, is not observed in our dataset, so we exclude the presence of both types of particles"***

MC3) What if there is a lot of ash particles and/or water droplets in the plume? Does this method work with ash- and droplet-free plumes (I seriously doubt if the second one exists in nature) only? If so, this statement should be revised: "This method is easily exportable to other volcanoes, to monitor magma extraction processes and the atmospheric sulphur cycle"

***If there is ash or pure water droplets (a possibility that we exclude for our specific case study), an ash and/or a pure water component must be added in Eq. 7, i.e. a second "aerosol component" of the plume extinction. In cases where ash/pure water droplets abundances are large, probably the SA signal would be lost. In any case, the limited information content of spectra (which is not estimated in this case, e.g. in terms of the degrees of freedom – DOF – following from Rodgers theory, which is the theoretical basis of our optimal-estimation inversion scheme) would be a***

*limiting factor for the simultaneous quantitative retrieval of SO2 + SA + ash/water. In presence of ash/water droplets, the inverse problem would be definitely more difficult. The application of our method to ashy plumes is an ongoing work. For the moment, as suggested by the Reviewer #1, we revised the sentence in the Abstract as follows: "This method is easily exportable to other volcanoes, to monitor magma extraction processes and the atmospheric sulphur cycle, in the case of ash-free plumes."*

MC4) The authors claim that the method is "easily exportable to other volcanoes" but have analyzed only 30 minutes of data from 2-years measurements conducted 2 decades ago. What is specific about the data that makes it "high quality spectra" and how likely is it to get such data elsewhere? This will elaborate on the requirements and limitations of the method. It would be great to see a second example that shows the applicability of the method to other eruptions/volcanoes.

*One thing that is required for our method, as it stands in this case study, is the "active nature" of the observations, i.e. using a lamp as a radiation source, which is not the general case for all archived data. The method can also be applied in the case of "passive" observations, i.e. using the sun as radiation source, but this would require an additional radiative transfer modelling to take into account the atmospheric path from sun to the FTIR spectrometer, through the plume. Second, the method would need to be slightly adapted in case of ashy plumes (see comment MC3). Last but not least, access to other past campaign data is not always public. For these reasons, unfortunately, we don't have, for the moment, a second case to show here. We plan to make dedicated campaigns, e.g. at Mt Etna (Italy), which is a second ideal scenario to apply our method as it is presented in this manuscript (usually little amount of ash, summit accessible for active observations, documented presence of sulphur compounds in gaseous and condensed phase). …and of course, we hope that other research teams will apply our method (or build upon this one to adapt to their specific situations) to their own measurements, which is a major motivation for our work and the existence of the present manuscript.*

MC5)- This MS is submitted to the special issue "StratoClim stratospheric and upper tropospheric processes for better climate predictions". Please explain how the results obtained from passively degassing volcano like Masaya (with tropospheric plumes) can be generalized and used for UTLS studies. Is it directly exportable to UTLS plumes? If not, what are the key aspects to consider?

*This method is directly exportable to UTLS plumes, as long as it is adapted to a different optical path and, possibly, observation geometry. For example, if satellite observations are concerned (as usually done for the observation of higher-altitude plumes and as in StratoClim project objectives) this method must be embedded into a full radiative transfer model. This requires considering a longer atmospheric path of the sensed radiation, up to the satellite, and the other radiatively interfering species present in the atmosphere. As for this latter aspect, please note that the careful wavelength selection carried out in the present work is a facilitating factor for the exportation of this method to that observation geometry. To position our work in this perspective, we added the following sentence at the very end of the conclusion section: "This method can also be exported to TIR satellite observations, like those from IASI (Infrared Atmospheric Sounding Interferometer) or others, as long as the full radiative transfer through the longer atmospheric path to the satellite platform is taken into account."*